# Multi-Boundary Empirical Path Loss Model for 433 MHz WSN in Agriculture Areas Using Fuzzy Linear Regression

**DOI:** 10.3390/s23073525

**Published:** 2023-03-28

**Authors:** Supachai Phaiboon, Pisit Phokharatkul

**Affiliations:** 1Department of Electrical Engineering, Faculty of Engineering, Mahidol University, 999 Salaya, Nakhorn Pathom 73170, Thailand; 2Department of Electrical Engineering and Energy Management, Faculty of Engineering, Kasem Bundit University, 1761 Phatthanakan 37 Alley, Suan Luang, Bangkok 10250, Thailand; 3Department of Computer Engineering, Faculty of Engineering, Mahidol University, 999 Salaya, Nakhorn Pathom 73170, Thailand

**Keywords:** open rural area, RSSI model, multi-boundary fuzzy linear regression, wireless sensor network, initial membership level

## Abstract

Path loss models are essential tools for estimating expected large-scale signal fading in a specific propagation environment during wireless sensor network (WSN) design and optimization. However, variations in the environment may result in prediction errors due to uncertainty caused by vegetation growth, random obstruction or climate change. This study explores the capability of multi-boundary fuzzy linear regression (MBFLR) to establish uncertainty relationships between related variables for path loss predictions of WSN in agricultural farming. Measurement campaigns along various routes in an agricultural area are conducted to obtain terrain profile data and path losses of radio signals transmitted at 433 MHz. Proposed models are fitted using measured data with “initial membership level” ($\mu_{AI}$). The boundaries are extended to cover the uncertainty of the received signal strength indicator (RSSI) and distance relationship. The uncertainty not captured in normal measurement datasets between transmitter and receiving nodes (e.g., tall grass, weed, and moving humans and/or animals) may cause low-quality signal or disconnectivity. The results show the possibility of RSSI data in MBFLR supported at an $\mu_{AI}$ of 0.4 with root mean square error (RMSE) of 0.8, 1.2, and 2.6 for short grass, tall grass, and people motion, respectively. Breakpoint optimization helps provide prediction accuracy when uncertainty occurs. The proposed model determines the suitable coverage for acceptable signal quality in all environmental situations.

## 1. Introduction

The applications of a wireless sensor network (WSN) have been extensively studied. The purpose of these applications is to facilitate data management as an intelligent network. Therefore, it is very important to study wave propagation between the receiver and transmitter. Substantial research has been performed on wave propagation between sensor nodes. In particular, WSNs for precision agriculture have grown and developed rapidly since green economy became a global trend. Precision agriculture aims to improve agricultural products, costs, and resource management. WSN technologies (e.g., long-range, low-power wide-area networks; Lora LPWAN) are remarkable devices for the wireless communication network in the Internet of Things (IoT) wireless applications for precision agriculture [1,2,3]. Solutions for long distance problems, low power consumption, and a low-cost communications network are necessary. For a WSN at 433 MHz, the radio wave propagates a longer distance than for a WSN at 2.4 GHz, although obstructions exist between the nodes in the communication areas. The WSN provides online monitoring and control, including data mining of crop growth and health through suitable node placement in the target area [4]. Therefore, the coverage and continued connectivity in the WSN are very important design factors for node placement in a specific area, which require suitable mathematic path loss models for certain environments.

There are many studies about the received signal strength indicator (RSSI) and path loss models in agricultural and forest areas. The RSSI studies focus on accurate positioning of WSNs in smart farming [5]. An empirical path loss model in forest environments at 2.4 GHz showed the predicted results of high grass–bushes, low grass, and bushes without grass at 65 m [6]. Rapidly growing weeds can cause problems with wireless communication. Therefore, the problem of wave propagation was studied in a grass area of a field. Accurate positioning was based on the received signal strength (RSS) and the angle of arrival at an agricultural farm (i.e., short and tall grass) [7]. Various changes in the farm environment lead to difficult problems in modeling optimal conditions. To solve this problem, the intelligent model can be trained to recognize different environments through machine learning (ML). Then, the ML path loss models can be studied for mobile communication and WSNs in agriculture because of the nonlinear property and uncertainty of path loss models. Artificial neural networks (ANNs) are used in ML by providing a learning environment. The ANNs for predicting the path loss of a macro cell showed accuracy compared to traditional models [8]. However, the path loss of the wave also depends on the terrain and frequency, which are very different. A ML-based urban canyon path loss prediction model based on extensive 28 GHz measurements from Manhattan achieved a prediction error (root–mean–square error; RMSE) of 4.8 ± 1.1 dB compared to 10.6 ± 4.4 dB and 6.5 ± 2.0 dB for the 3GPP line of sight (LOS) prediction [9]. There are many different architectures of neural networks used to model path loss prediction, such as a convolutional neural network for fixed wireless access in suburban areas, obtaining an 8.59 dB RMSE [10]. An ANN-based path loss prediction for the wireless communication network at 2.5 GHz presented the rules for training and accuracy prediction [11]. The ANN and the adaptive neuro fuzzy interference system were studied to build very high frequency and ultra-high frequency path loss models. These models showed better prediction compared with selected empirical models and field measured data [12]. Lastly, path loss prediction based on ANN and the measurements at 189.25 MHz and 479.25 MHz provided the best ANN parameters and minimal error [13].

The abovementioned studies showed that RSSI varies according to environment and uncertainty, such as large-scale distance change, antenna height, tall grass, weeds, and blockages between nodes. These make the RSSI fade, resulting in delayed or low-quality communications. The present study applies the fuzzy theory to create a model for path loss prediction which corresponds to the nature of the RSSI uncertainty in different environments. This model is referred to herein as “multi-boundary fuzzy linear regression” (MBFLR). It consists of a center line, two upper boundary lines, and two lower boundary lines that cover almost all the RSSIs in the coverage areas. These components make the proposed model provide limited interference in the upper boundary and lowest RSSI for interruption. The proposed model defines the WSN characteristic in an agricultural farm through ML. Moreover, it can predict the RSSI under uncertainty fading with accuracy and can be applied to similar areas.

The following contributions are made to address the previously indicated issues:(1)A ML path loss model with RSSI fluctuation boundaries is proposed using MBFLR based on the measurements in different environments for short grass, tall grass, weeds, and blockage.(2)A breakpoint distance optimization is proposed for accurate prediction.(3)The measured RSSI data are captured using Lora LPWAN at 433 MHz for different environments.

The remainder of this paper is structured as follows: Section 2 presents the related work; Section 3 proposes the MBFLR model; Section 4 demonstrates the experimental setup; the results are depicted in Section 5; the discussion and description are in Section 6; and finally, the conclusion and future work are presented in Section 7.

## 2. Related Work

Many research studies have focused on propagation path loss models for vegetation at frequencies of 200 MHz to 95 GHz within a distance of 400 m, such as in a review by Dessales et al. in [14]. Studies on radio propagation were performed on short and tall grass environments to support animal grazing in large-scale farming at a frequency of 2.4 GHz by Olasupo et al. in [15] and Alsayyari et al. in [16]. Wireless communication near the ground was proposed with the plane–earth model for VHF and UHF bands by Meng et al. in [17]. The researchers limited their interest to specific phenomena, such as the impact of near-ground or surface components on the signal propagation in different environments. Additionally, path loss models with breakpoint distance on-ground, near-ground and aboveground (5 cm, 50 cm, and 1 m) were measured and analyzed at a frequency of 470 MHz by Tang et al. in [18]. The path loss models for WSN in a palm garden used a near-ground LOS model and tree attenuation factors by Anzum et al. in [19]. A path loss model on a grass field was proposed at WiFi 2.4 GHz by García et al. in [20]. Additionally, the effects of people motion on WSN 433 MHz and 868 MHz in building monitoring were analyzed by Dessales et al. in [21]. In addition, a fuzzy system has been used in path loss modeling. The three input fuzzy membership functions, namely, residual power of sensor nodes, distance of the node from the base station node, and RSSI, together with two output fuzzy membership functions, focused on the influence of the RSS on the cluster head nodes and rounds [22]. Lastly, AI with ANFIS (adaptive neuro fuzzy inference system) for near-ground WSNs in forest environments were proposed by Hakim et al. in [23]. This ML path loss model provided the lowest RMSE at 433 MHz for an open dirt road environment compared with the empirical models such as optimized FITU-R near ground model, Okumura-Hata model, and ITU-R maximum attenuation free space model.

In this study, propagation models in the form of RSSI–distance relationships at 433 MHz for WSN at antenna heights of 0.2 m, 0.7 m, and 1.2 m are proposed. Furthermore, we apply fuzzy linear regression (FLR) with “initial membership level” for fitting RSSI data in a short-grass field. The results also cover RSSI data in long grass and with people motion within extended boundaries.

## 3. Multi-Boundary Fuzzy LR

The relationship between the RSSI and the distance (d) for the radio wave propagation is expressed as follows in a mathematical LR form:(1)$${RSSI}_{i}=A-Blog\left(d_{i}\right)$$
where *A* and *B* are the relationship coefficients. The distance and the *RSSI* values express the regression coefficients with fuzzy numbers as follows due to the wave propagation uncertainty:(2)$${RSSI}_{i}^{\prime }=\dot{A}-\dot{B}log\left(d_{i}\right)$$

The fuzzy membership functions of the fuzzy *RSSI* variables (*RSSI’*) can be derived from the measurement uncertainties. The membership functions of fuzzy coefficients $\dot{A}$ and $\dot{B}$ can be evaluated using the fuzzy regression analysis based on the fuzzy extension principle [24,25,26]. The left–right (L–R) presentation of the fuzzy number provides a suitable means for representing the fuzzy coefficients. Let $\dot{A}$ be the coefficient expressed as follows:
(3)$$\dot{A}=f(C,L,R)$$
where *C* is the central value, and *L* and *R* are the left and right spreads, respectively. The membership function $\mu_{A}(x)$, of the triangular *L*-*R* fuzzy number is given by Equation (4).
(4)$$\mu_{A}\left(x\right)=\left\{\begin{array}{c} 1-\frac{C-x}{L}forC-L\leq x\leq C,L>0\\ 1-\frac{x-C}{R}forC\leq x\leq C+R,R>0 \end{array}\right.$$

Note that the maximum *l-r* spreads are the boundary of the model that occurs at membership level 0.0. We incorporate the uncertainty not captured in the available measurement data sets by using the proposed MBFLR model and define a variable membership level $\mu_{A}$ as a factor [27] for fitting the certainty captured by RSSI data, called the “initial membership level” ($\mu_{AI}$), to extend the L–R spreads for the uncertainty not captured at the lower membership levels (see Figure 1). According to this approach, each of the measured RSSI must be within the boundaries around the estimated regression curves at lower $\mu_{AI}$ values. The spread of the membership function and the fuzziness of the regression variables can be controlled by specifying the $\mu_{AI}$ level between 0 and 1. The $\mu_{AI}$ value is useful in considering quantified uncertainties, such as the maximum or minimum RSSI, caused by the fading electromagnetic wave. Accordingly, the left and right spreads of the L–R fuzzy numbers $\dot{A}$ and $\dot{B}$ can be expressed as follows:(5)$$C_{A}-L_{A}\;(1{-\mu }_{AI})\leq \;\dot{A}\;\leq C_{A}+R_{A}\;(1{-\mu }_{AI})$$
(6)$$C_{B}-L_{B}\;(1{-\mu }_{AI})\leq \;\dot{B}\;\leq C_{B}+R_{B}\;(1{-\mu }_{AI})$$

The spread of the RSSI fuzzy number RSSI’ obtained from the measurement uncertainty analysis is expressed as follows:(7)$$C_{RSSI}{-L}_{RSSI}{\leq C}_{RSSI}\leq C_{RSSI}+R_{RSSI}$$

Based on the uncertainty RSSI measurement, the fuzzy membership function of the output has a triangular shape (see Figure 1). The two corners of the triangular base represent the zero-membership level of fuzzy set A. For example, an initial membership level ($\mu_{AI}$) of 0.4 provides the lower bound of the fuzzy data curve 0.0(L) and 0.2(L) that will intersect with the left spread of the lower membership values 0.0 and 0.2, respectively. Similarly, the higher upper bound of the fuzzy rating curve 0.0(U) and 0.2(U) will intersect with the right spread of the lower membership values 0.0 and 0.2, respectively. Therefore, Equations (2) and (5)–(7) can be combined, such that the lower bound of the fuzzy data curve intersects at the boundary of the left spread, and the upper bound intersects at that of the right spread. Therefore, the lower and upper bounds of the fuzzy regression curve can be presented in the following forms:(8)$$\{C_{a}-L_{a}(1-\mu_{AI})\}-\{C_{b}-L_{b}(1-\mu_{AI})\}log\left(d_{i}\right)\geq \;C_{{RSSI}_{i}}-L_{{RSSI}_{i}}$$
(9)$$\{C_{a}+R_{a}(1-\mu_{AI})\}-\{C_{b}+R_{b}(1-\mu_{AI})\}log\left(d_{i}\right)\;\leq \;C_{{RSSI}_{i}}+R_{{RSSI}_{i}}$$

As a function of the distance, the RSSI has two distinct regions due to the first Fresnel zone region, that is, before and after the breakpoint distance. The RSSI is separated as follows at the breakpoint distance $d_{bp}$ in Equation (10):(10)$${RSSI}_{i}^{\prime }={\dot{A}}_{1}-{\dot{B}}_{1}log\left(d_{i}\right)\;\mathrm{for}\;d_{i}\;\leq \;d_{bp}$$
(11)$${RSSI}_{i}^{\prime }={\dot{A}}_{2}-{\dot{B}}_{2}log\left(d_{i}\right)\;\mathrm{for}\;d_{i}\;>\;d_{bp}$$
(12)$${\dot{A}}_{1}-{\dot{B}}_{1}log\left(d_{i}\right)={\dot{A}}_{2}-{\dot{B}}_{2}log\left(d_{i}\right)\;\mathrm{for}\;d_{i}\;=\;d_{bp}$$
where subscripts 1 and 2 denote the before and after breakpoint ranges, respectively. The consideration of the two curves meeting at the breakpoint $d_{bp}$ leads to the expression of Equations (13)–(15):(13)$$\left.\begin{array}{c} \{C_{a1}-L_{a1}(1-\mu_{AI})\}-\{C_{b1}-L_{b1}(1-\mu_{AI})\}log\left(d_{i}\right)\leq C_{{RSSI}_{i}}-L_{{RSSI}_{i}}\\ \left\{C_{a1}+R_{a1}\left(1-\mu_{AI}\right)\right\}-\left\{C_{b1}+R_{b1}\left(1-\mu_{AI}\right)\right\}log\left(d_{i}\right)>C_{{RSSI}_{i}}+R_{{RSSI}_{i}} \end{array}\right\}\;\mathrm{for}\;d_{i}\;\leq \;d_{bp}$$
(14)$$\left.\begin{array}{c} \{C_{a2}-L_{a2}(1-\mu_{AI})\}-\{C_{b2}-L_{b2}(1-\mu_{AI})\}log\left(d_{i}\right)\leq C_{{RSSI}_{i}}-L_{{RSSI}_{i}}\\ \left\{C_{a2}+R_{a2}\left(1-\mu_{AI}\right)\right\}-\left\{C_{b2}+R_{b2}\left(1-\mu_{AI}\right)\right\}log\left(d_{i}\right)>C_{{RSSI}_{i}}+R_{{RSSI}_{i}} \end{array}\right\}\mathrm{for}\;d_{i}\;>\;d_{bp}$$
(15)$$\left.\begin{array}{c} \left\{C_{a1}-L_{a1}\left(1-\mu_{AI}\right)\right\}-\left\{C_{b1}-L_{b1}\left(1-\mu_{AI}\right)\right\}log\left(d_{i}\right)=\\ \{C_{a2}-L_{a2}(1-\mu_{AI})\}-\{C_{b2}-L_{b2}(1-\mu_{AI})\}log\left(d_{i}\right)\\ \left\{C_{a1}+R_{a1}\left(1-\mu_{AI}\right)\right\}-\left\{C_{b1}+R_{b1}\left(1-\mu_{AI}\right)\right\}log\left(d_{i}\right)=\\ \left\{C_{a2}+R_{a2}\left(1-\mu_{AI}\right)\right\}-\left\{C_{b2}+R_{b2}\left(1-\mu_{AI}\right)\right\}log\left(d_{i}\right)\\ C_{a1}-C_{b1}log\left(d_{i}\right)=C_{a2}-C_{b2}log\left(d_{i}\right) \end{array}\right\}\mathrm{for}\;d_{i}\;=\;d_{bp}$$

The minimum spread criteria are considered to evaluate the fuzziness output. The minimum spread of the fuzzy numbers is obtained by minimizing the output support for a total of n observations consisting of m distance observations:(16)$$v=\overset{p}{\underset{i=1}{\sum }}|\left\{C_{a1}+R_{a1}\;\left(1-\mu_{AI}\right)\right\}-\left\{C_{b1}+R_{b1}\left(1-\mu_{AI}\right)\right\}log\left(d_{i}\right)\;-\left\{C_{a1}-L_{a1}\;\left(1-\mu_{AI}\right)\right\}-\left\{C_{b1}-L_{b1}\left(1-\mu_{AI}\right)\right\}log\left(d_{i}\right)|\;+\overset{\begin{array}{c} m \end{array}}{\underset{i=p}{\sum }}|\;\left\{C_{a2}+R_{a2}\;\left(1-\mu_{AI}\right)\right\}-\left\{C_{b2}+R_{b2}\left(1-\mu_{AI}\right)\right\}log\left(d_{i}\right)\;-\left\{C_{a2}-L_{a2}\;\left(1-\mu_{AI}\right)\right\}-\left\{C_{b2}-L_{b2}\left(1-\mu_{AI}\right)\right\}log\left(d_{i}\right)|$$

The set of Equations (10)–(16) provides a mathematical formulation of the fuzzy regression analysis problem using the fuzzy form of the input and output variables. The formulation leads to an optimization problem for coefficient evaluation in Equations (8) and (9) in terms of the central value and the left and right spreads. Equations (13) and (14) provide linear inequality constraints, while Equation (15) provides an equality constraint for optimization. Figure 2 illustrates the step-by-step procedure for the fuzzy regression.

## 4. Experimental Setup

The propagation model for the short and tall grass was investigated by performing an extensive experiment campaign using the LoRa 433 MHz WSN at a smart farm in Nakornchaisri. The smart farm located in the West suburbs of Bangkok has GPS coordinates of 13.77242 and 100.17602. The LoRa device is a non-cellular, long-range and low-power wireless technology widely used in the IoT to communicate between nodes and gateways. Table 1 presents the WSN specification and measurement parameters. The equipment characteristics influenced the RSSI (e.g., spreading factor (SF) and bandwidth) [19]. The first measurement step was conducted in normal short grass. The other measurements were implemented for the uncertainty’s phenomena of the surrounding environments. The measurement showed varying RSS data for each antenna height. Natural propagation phenomena known as the diffraction, refraction, and reflection of the transmitted signal caused by the surroundings of the rural open environment were also observed. Some of the excess attenuations were possibly caused by the uncertainty’s phenomena, such as the grass height, weeds, uneven ground, and animal movement. The measurements were performed in both the dry and rain seasons; however, the effects of rainfall for frequencies below 10 GHz showed insignificant effects on the signal propagation [28]. The SF and bandwidth were fixed to 7 and 125 kHz, respectively, for the distance between 0 m and 40 m to concentrate on the signal fluctuation in the interesting surroundings.

### 4.1. Short Grass

The experiment was conducted in a corridor grass area surrounded with banana and mango trees. The WSN comprised fixed (receiving node or Rx) and mobile (transmitting node or Tx) nodes (please see Figure 3 (left)). The Tx mobile node was moved every 5 m along a straight line between the Tx and Rx nodes. The fixed node used the collected RSSI via a notebook to estimate the path loss between the two nodes based on MBFLR. The grass area was 5.0 m × 100 m with the minimum and maximum distances of 1 and 40 m, respectively. The measurements were performed in both forward and reverse directions during the months of January to June 2022. The RSSI values were collected for approximately 60 s for each measured point and repeated thrice to obtain 540 RSSI data sets for each antenna height. The antenna heights for the transmitting and receiving nodes were 0.2 m, 0.7 m, and 1.2 m above the ground. The measurements were repeated every month from no grass (~0 cm) to tall grass (~50 cm). We also considered the breakpoint distance to separate the propagation region into two zones for our measurement. Considering the first Fresnel zone, the breakpoint distance depended on the antenna heights above the ground, as shown in Equation (17):(17)$$d_{bp}=\frac{4h_{t}h_{r}}{\lambda }$$
where $h_{t}$ and $h_{r}$ are the transmitting and receiving antenna heights, respectively, and λ is the carrier wavelength. Figure 4 depicts the RSSI–distance relationship of the large-scale fading for the distance before and after the breakpoint. Note that $d_{bp}$ with a 1.2 m antenna height was at an 8.3 m (0.92 dB) distance from the receiving node for modeling. The PLEs before $d_{bp}$ were approximately 20 (19.4), while the PLEs after $d_{bp}$ were more than 20 (25.87) for the LOS propagation theory [29].

### 4.2. Tall Grass and Weeds

The uncertainty’s phenomena in the tall grass and weed between the Tx and Rx nodes were measured to capture the impossible data in the short grass environment. The tall grass and weed height was approximately 30–50 cm. Tall grass and weed covered the surface of the measured area by approximately 70%. The measurements were performed with a large-scale fading both before and after the breakpoint distance. The measured RSSI was used to model and validate MBFLR. Figure 5 shows the uncertainty in the observed RSSI–distance relationship. The PLEs before $d_{bp}$ were approximately 20 (18.37) like the short grass, while those after $d_{bp}$ were more than 26.95 (i.e., more than the short grass (25.87)) because of the blockage from the tall grass at 0.2 m antenna height above the ground.

### 4.3. People Motion

The effect of people moving between the communication nodes at the 433 MHz influenced the received signal quality, such as in building communication [21]. Therefore, this experimental study was performed in two scenarios, that is, random and full blockage, to detect and capture the effects. For the random blockage, one farmer continually walked between the two nodes in the experiment, while for the full blockage, two farmers stood near two nodes (see Figure 6). Figure 7 and Figure 8 illustrate the effect results for the first and second scenarios, respectively. Note that the PLE before $d_{bp}$ was less than 20, and the minimum RSSI occurred at a 1 m distance due to the fast fading from the obstruction (i.e., people moving), while the PLE after $d_{bp}$ was still more than 20. Note that the blockages placed the RSSI in a lower boundary.

## 5. Results

The measurement results showed a varying signal strength for each measurement point and surrounding environment. Figure 4, Figure 5, Figure 6, Figure 7 and Figure 8 depict a few interesting facts. First, the PLEs before $d_{bp}$ were still approximately 20 from the Fresnel zone clearance effect (see Figure 4 and Figure 5) because the transmitter and receiver antenna heights were above the grass and weed surfaces. In general, the diffraction loss may be neglected if an obstruction does not block the volume contained within the first Fresnel zone [30]. However, in the case of the blockage between the nodes in Figure 6, the RSSIs were in low levels and made the PLEs lower than 20 because of the Fresnel zone non-clearance effect. According to this rule of thumb, as long as 55% of the first Fresnel zone is not kept clear, a further Fresnel zone clearance will significantly alter the diffraction loss. However, the PLEs were more than 20 after $d_{bp}$, confirming the Fresnel zone theory. Another interesting fact was that the large fluctuation occurred at all distances (see Figure 7) in the case of one blockage moving between the nodes (see Figure 6a). This was because the Fresnel zone clearance was changed by the blockage movement. In contrast, in the case of the blockages fixed at each node (see Figure 6b), large attenuation and fluctuation occurred at only a 1 m distance and small fluctuation occurred in the distance between 5 and 40 m intervals (see Figure 8). The measured RSSIs were fitted to the models and compared as shown below.

### 5.1. Proposed LR Models

Figure 4 and Equations (18) and (19) present the empirical propagation models with the separated distances for the short grass environment.
(18)$$RSSI_{1}\left(d\right)=-53.79-19.42\log\left(d\right);\;d\;\leq \;d_{bp}$$
(19)$$RSSI_{2}\left(d\right)=-49.31-25.87\log\left(d\right);\;d\;>\;d_{bp}$$
where d is the distance between the two nodes. The RSSI models were the CI (close in free space) and dual-slope models. The PLEs before the breakpoint distance were approximately 20, while those after were more than 20. These values confirmed the wave propagation theory for the LOS condition. We used the breakpoint distance at 15 m (log (15) = 1.18) (Figure 4) calculated from Equation (11) with a 1.2 m antenna height to determine the overall separated parts of communication. Similarly, the proposed models for the tall grass and weed environments are presented as follows:(20)$$RSSI_{1}\left(d\right)=-56.04-18.37\log\left(d\right);\;d\;\leq \;d_{bp}$$
(21)$$RSSI_{2}\left(d\right)=-43.76-26.95\log\left(d\right);\;d\;>\;d_{bp}$$

Equation (20) shows that the PLE is mostly unchanged compared to Equation (18). This follows from the theory that the PLE is still approximately 20 in the first Fresnel zone region for the incomplete blocking case in tall grass and weed. However, this effect occurs after the breakpoint and makes the PLE increase as shown in Figure 5 and Equation (21).

For the case of people moving, the experiment was performed in two scenarios: (1) random blocking, and (2) full blocking. For the first scenario, one farmer walks forward and reverse between the communication nodes for all antenna heights as shown in Figure 6a. The propagation result is shown in Figure 6b and expressed in Equations (22) and (23),
(22)$$RSSI_{1}\left(d\right)=-65.00-8.95\log\left(d\right);\;d\;\leq \;d_{bp}$$
(23)$$RSSI_{2}\left(d\right)=-43.76-23.52\log\left(d\right);\;d\;>\;d_{bp}$$

Equation (22) shows that the PLE before the breakpoint was less than 20 since the random movements make the RSSIs fall, especially at the reference distance (1 m) as shown in Figure 7. In the case of distance before breakpoint, the maximum radius of the first Fresnel zone is small compared with the obstruction side. This makes the path loss slope decrease greatly. In the case of distance after breakpoint, the radius is very large compared with the obstruction side; therefore, the path loss slope shows almost no change. Thus, breakpoint distance optimization makes the prediction accurate, especially in the case of the distance after breakpoint. Thus, the PLE in Equation (23) was still more than 20. In the case of the second scenario, two farmers fully blocked the radio wave at each node (see Figure 6b). Figure 8 depicts the propagation result expressed in Equations (24) and (25),
(24)$$RSSI_{1}\left(d\right)=-70.47-10.14\log\left(d\right);\;d\;\leq \;d_{bp}$$
(25)$$RSSI_{2}\left(d\right)=-59.67-21.84\log\left(d\right);\;d\;>\;d_{bp}$$

In Figure 8, the RSSI in each distance was collected in the lower bound because the radio paths were blocked from obstructions. This caused a small effect from the multi-path fading. Equation (24) shows that the PLE before the breakpoint was mostly equal to that in Equation (22) with one moving person and that with the PLE of Equation (25), which was also more than 20.

### 5.2. Proposed MBFLR Models

The LR model above cannot provide the minimum RSSI that influences disconnected communication, maximum RSSI, and interference to another network. Hence, MBFLR was applied. The individual RSSI measurements with all the antenna heights (0.2 m–1.2 m) were fuzzified and aggregated into a combined uncertainty for the RSSI–distance relationship based on the LOS short grass environment conditions. Two different groups of uncertainty in the RSSI measurements were considered before and after the breakpoint distances. The RSSI uncertainty was expressed herein as triangular fuzzy numbers with a membership level of 0 to 1. The fuzzy aggregation of the uncertainties was used in the fuzzy LR analysis with fuzzy output variables. For simplicity, the symmetrical triangular L–R fuzzy numbers for coefficients A1, B1, A2, and B2 (Equation (8) were equal to the left and right spreads, respectively. The decision variables for the fuzzy regression were reduced to eight, that is, central values Ca1, Ca2, Cb1, and Cb2 and spreads Ra1, Ra2, Rb1, and Rb2 for the distance before and after the breakpoint distance coefficients. An initial membership level ($\mu_{AI}$) was used for the modeling, which increased or decreased the spread of the fuzzy regression curve and the output spread. The uncertainty in the RSSI measurements was expressed by the wide spread of their fuzzy numbers; thus, the $\mu_{AI}$ values can be varied for optimization. The fuzzy regression analysis result using the minimum spread criteria (Equation (16)) with 0.4 $\mu_{AI}$ is expressed as:(26)$$RSSI_{1}\left(d\right)=\left[-55.4,-18.2\right]+\left[-16.2,\;0.2\right]\log\left(d\right);\;d\;\leq \;d_{bp}$$
(27)$$RSSI_{2}\left(d\right)=\left[-42.9,-22.5\right]+\left[-26.8,\;4.9\right]\log\left(d\right);\;d\;>\;d_{bp}$$
where the central and spread of fuzzy coefficients $\dot{A}$ and $\dot{B}$ are −55.4, −18.2 and −16.2, 0.2 respectively for Equation (26), and are −42.9, −22.5 and −26.8, 4.9 respectively for Equation (27). The analysis produced upper (U) and lower (L) curves bounding the fuzzy distance and RSSI data. The uncertainty bound curves for the different membership levels represented $\mu_{AI}$ of belonging of the RSSI values corresponding to a particular measured distance. The closer the membership level was to one, the wider the boundaries were. The spread of the fuzzy regression curves depended on the $\mu_{AI}$ used during the regression analysis, while that of the uncertainty bound curves depended on the $\mu_{AI}$ of 0.4 and two different levels of belonging (i.e., 0.2 and 0.0) for the measured RSSI in different grass environments. The curves between the lower 0.4(L) and upper 0.4(U) bounds represent the total uncertainty of the RSSI–distance relationship in the normal short grass area (see Figure 9). Meanwhile, those between 0.2(L) and 0.2(U) represent the uncertainty in the short and tall grass and weed (see Figure 10). Similarly, the curves between 0.0 L and 0.0 U covered all the uncertainties in the available data set of the short and tall grass uncertainties, including the effect of the one moving person (see Figure 11). However, the RSSI fell to the lower bound (0.0 L) in the case with two people blocking because the radio waves were mostly blocked at both nodes (see Figure 12). This case cannot influence the WSN disconnection because of the limited low signal (>−91 dB) and very large change of signal. Almost all RSSIs were between the central line (1.0) and the lower boundary (0.0 L) (see Figure 13).

### 5.3. Model Comparison

The RSSI data were collected herein for 433 MHz LoRa channels, and then analyzed. The data comparison was then used to represent the path loss models’ behavior. To compare our RSSI and path loss models, the received signals were obtained through Equation (28):(28)$$RSSI\left(dB\right)=Pt+Gt+Gr-Path\;loss$$
where Pt is the transmitting power (*dB*), and Gt and Gr are the gains of the transmitting and receiving antennas, respectively.

The RMSE was calculated as follows to observe the deviation of the measurement and related empirical models:(29)$$RMSE=\sqrt{\frac{\sum_{i=1}^{N}{\left(\mathrm{Measured}\;RSSI-\mathrm{Model}\right)}^{2}}{N}}$$

In this work, we attempt to provide different perspectives. Therefore, we assessed the performance by comparing the proposed model with the three following propagation models of a similar environment at the same frequency, namely: One Slope Path Loss Prediction Model (433 MHz), Optimized FITU-R NGF, and ITU-R MA FSPL.

(1)One-slope Path Loss Prediction Model (433 MHz)

An empirical path model for a palm plantation was proposed based on the measurement of the LoRa 433 MHz at the available spreading factors (SF7–SF12) and bandwidths (125, 250, and 500 kHz) for wireless sensor networks [19]. The model for the LOS with grass at the 1 m antenna height is presented below:(30)$$PL_{LOS\left(433MHz\right)}=PL_{0}+10nlog\left(d\right)\;$$
where $PL_{0}$ is the reference path loss at the 1 m distance from the transmitter; d is the distance between the transmitter and the receiver; and n is the PLE equal to 2.37 for SF7 and BW 125 kHz.

(2)Optimized FITU-R Model for Near-ground Forest (Optimized FITU-R NGF)

This model proposes the optimization of the FITU-R (fitted ITU-R) model for the near-ground path loss modeling in a forest environment [17]. It considers the plane earth model that explains the direct ray in addition to the ground-reflected ray received by the receiver in Equation (31).
(31)$$PL_{PlaneEarth}\left(db\right)=\left(40\;log\;d\right)-\left(20\;log\;ht\right)-\left(20\;log\;hr\right)$$

The ITU-R foliage attenuation (AITU−R foliage) was added into the plane earth model as follows:(32)$$AITU-R{\;}_{foliage}\left(dB\right)+PL_{PlaneEarth}\left(dB\right)=0.2\;f^{0.3}d^{0.6}+\left(40\;log\;d\right)-\left(20\;log\;ht\right)-\left(20\;log\;hr\right)$$
where *f* is the frequency carrier in GHz; *ht* is the antenna transmitter height in meters; *hr* is the antenna receiver height in meters; and *d* is the distance between the transmitter and the receiver in meters.

(3)ITU-R Maximum Attenuation and Free Space Path Loss (ITU-R MA FSPL)

This model is recommended by the International Telecommunications Union for the 30 MHz–30 GHz range. In the case of the LOS, the ground-reflected ray in this environment is negligible because the forest ground is covered with full grass and/or weed that absorbs the radio waves. However, in the case of the NLOS, the maximum attenuation is calculated and combined with the LOS, as shown in Equation (33) [31]:(33)$$PL_{MA-ITU-R}\left(dB\right)=A_{M}\left(1-e^{-Rd/A_{M}}\right)+32.44+20\;log\;d+20\;log\;f\;$$
where $A_{M}$ is the maximum excess attenuation in *dB*; *R* is the initial slope of the attenuation curve in *dB*/m; *d* is the distance between the transmitter and the receiver in km; and *f* is the frequency carrier in MHz.

## 6. Discussion

To optimize the MBFLR model for all the measured RSSIs, the measured RSSI data were first fitted with the LR dual-slope models (Equations (18) and (19)) for the short grass environments. All measured RSSIs were in the boundary with membership level 0.4 (see Figure 9). Meanwhile, in the case of the tall grass and weed in Equations (20) and (21), the boundary was expanded with membership level 0.2 to cover the measured RSSI (see Figure 10). The central line of both graphs shows the same equation; however, the lowest RSSI occurred at a 40 m distance with a −103 dB value when the environment was tall grass and weed. In the case of one blockage movement in Equations (22) and (23), the measured RSSIs fluctuated along the measurement distances of 1–40 m with a maximum range of 28 dB (−66 to value −94) at a 10 m distance (see Figure 11). The third boundary was also expanded with membership level 0.0 to cover all the measured RSSIs. However, outward RSSIs existed at the lower boundary, which occurred only at the 1 m distance with a maximum range of 23 dB (−68 to −91) when the two blockages were close to the transmitting and receiving nodes (see Figure 12). This model is shown in Equations (24) and (25). Note that the PLEs before $d_{bp}$ in the case of the blockages were small (i.e., less than 20) due to the weak RSSI at the reference distance of 1 m. This phenomenon can hardly occur because the blockage was close to the monitoring node (i.e., receiving node), which should be clear from any obstructions. In addition, this minimum signal level (−91) was sufficient to connect the communication. Finally, the MBFLR model can be expressed in Equations (26) and (27) with a 0.4 $\mu_{AI}$ value, which can predict the path loss when the environment changes due to natural and human causes. This makes the WSN more stable after planning with the proposed model.

In the statistical error analysis results presented in Table 2, the smallest value indicated that the model had the best matching performance between the predicted and observed values. Based on the statistical evaluation, the proposed MBFLR model had the lowest RMSE scores of 0.8, 1.2, and 2.6 for the short and tall grass and blockages, respectively. Note that the observed values were compared with the predicted boundary values. The proposed dual-slope models provided a large RMSE because of the large RSSI fluctuation from the regression line, especially in the tall grass and weed environments (see Figure 10). In the case of the LR line models, the best model was FITU-R MA FSPL, with the lowest RMSE score of 4.08 for the short grass environment. However, the RMSE scores for the tall grass and blockage environments were 10.9 and 8.1, respectively, due to the large RSSI fluctuation. The second-best model was the One Slope PL_LOS 433 MHz model, which showed the lowest RMSE score of 5.7 for the short grass environment. The RMSE scores for the tall grass and blockage environments were 11.8 and 8.3, respectively. The worst model was the Optimized FITU-R NG, with the lowest RMSE scores of 25.2, 27.7, and 24.7 for the short and tall grass and blockage environments, respectively, especially at the distance before $d_{bp}$. However, this model meets the upper boundary and the lower boundary of the MBFLR model at distance of approximately 1 m and 40 m, respectively (see Figure 14).

## 7. Conclusions

In this study, we investigated and analyzed the RSSI behavior on grass environments with low transmitter and receiver antenna heights from the ground. Accordingly, we introduced a fuzzy MBFLR model to predict the near-ground path loss in different environment scenarios. The obtained results showed that the proposed MBRLR model with multi-boundaries provides the best RMSE with measurement results in the short grass, tall grass and weed, and blockage environments. The fuzzy MBFLR model achieved the lowest RMSE score of 0.8 for the short grass environment, the lowest RMSE score of 1.2 for the tall grass with weed environment, and an RMSE score of 2.6 for one and two blockages between the nodes. The Optimized FITU-R Near-ground Model achieved the highest RMSE score of 25.2 for the short grass environment, the lowest RMSE score of 24.7 for the tall grass with weed environment, and an RMSE score of 27.7 for one and two blockages between the nodes. A large error was also found for the NLOS in the case of blockages, although this model was used for the vegetation loss. The ITU-R MA FSPL, One Slope PL open-area 433 MHz, and dual-slope models based on the measurements provided good predictions only for the open-area environment without any blockages. The RMSE values indicated that the Optimized FITU-R Near-ground Model was not suitable for predicting the RSSI value in all situations in the grass environment due to its large PLE. By contrast, the fuzzy MBFLR performed the most accurate prediction and cover of the RSSI values for the near-ground propagation for all environments. However, further studies should be carried out on a large scale in other environments or with different types of vegetation and performed during all seasons in order to obtain coverage path loss predictions as well.

## Figures and Tables

**Figure 1 sensors-23-03525-f001:**
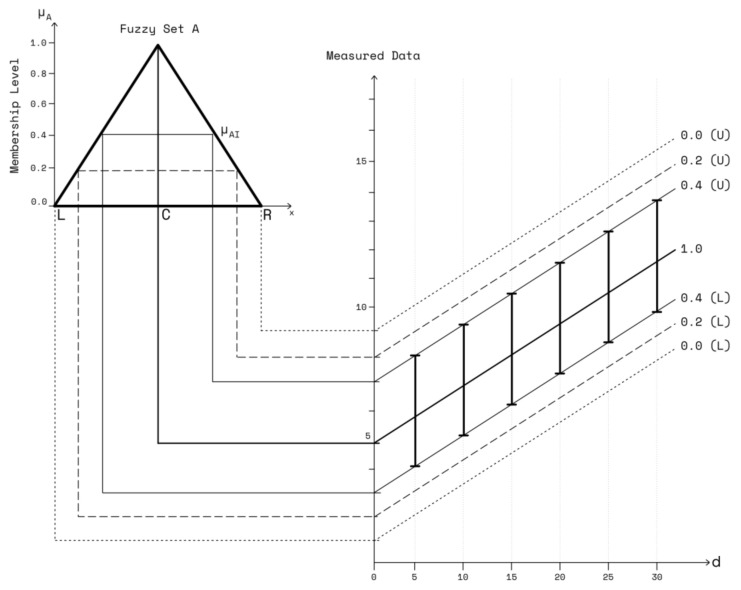
Concept of MBFLR.

**Figure 2 sensors-23-03525-f002:**
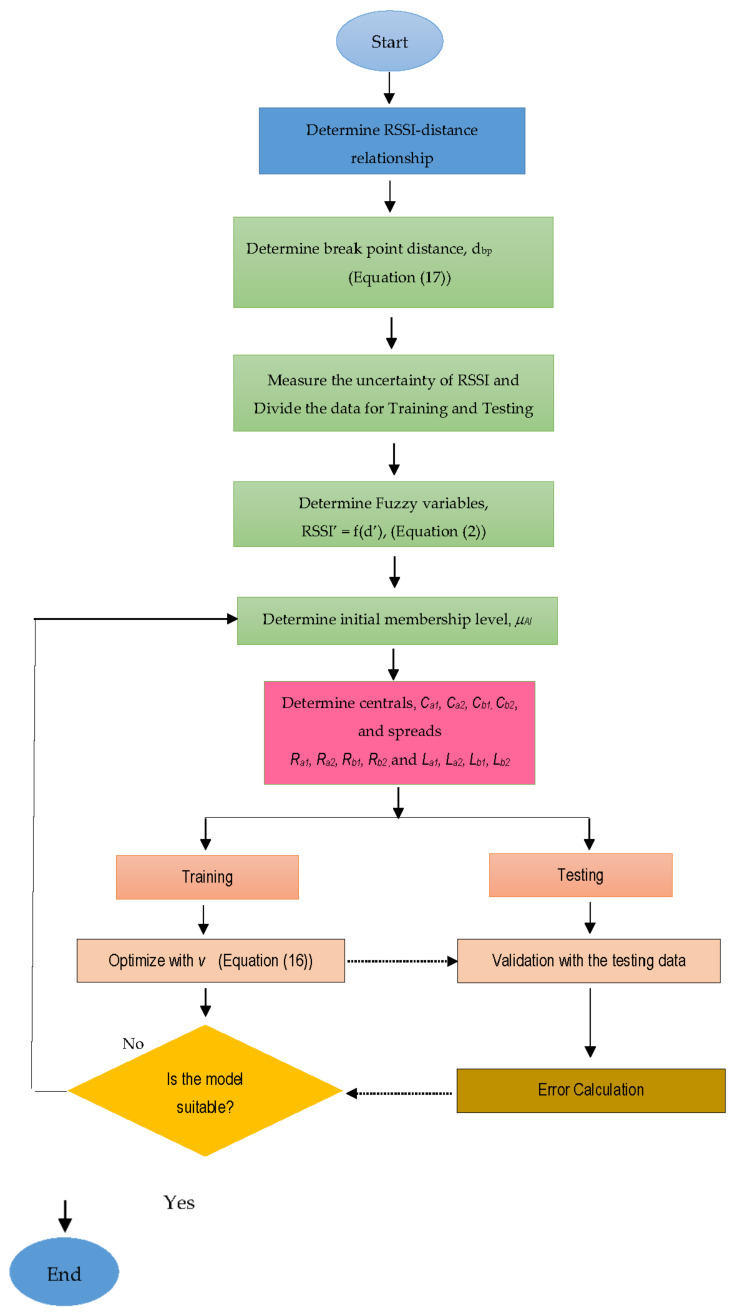
Flow chart of MBFLR fitting.

**Figure 3 sensors-23-03525-f003:**
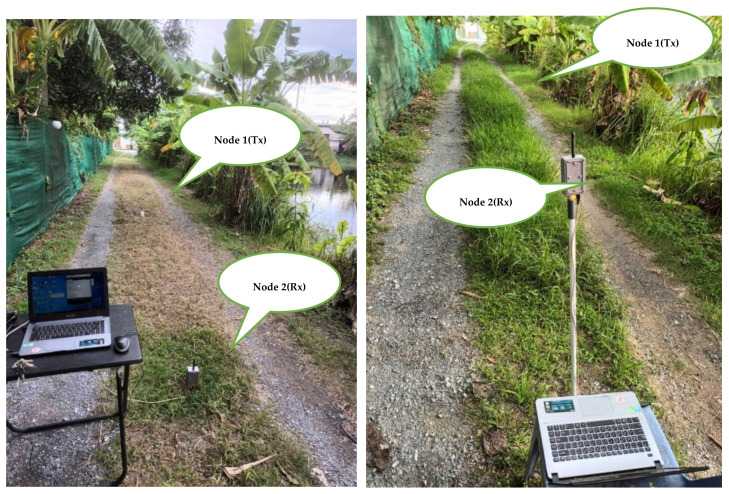
RSSI measurement for short grass (left) and tall grass (right).

**Figure 4 sensors-23-03525-f004:**
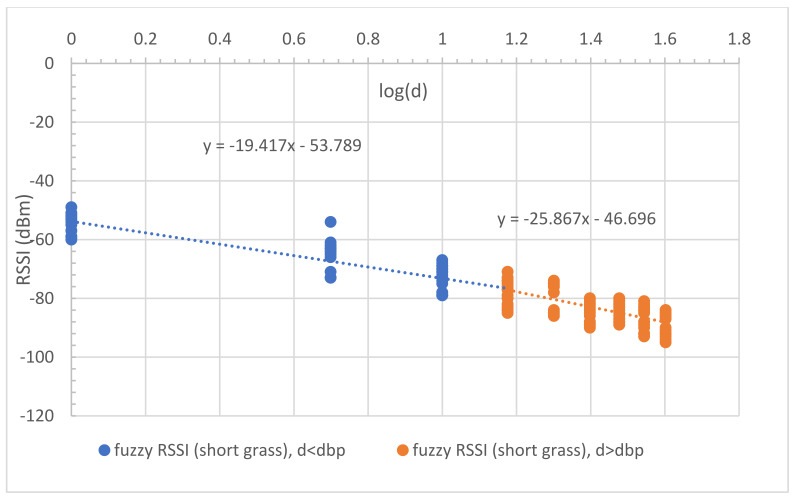
Uncertainty in observed RSSI and distance of short grass, for all antenna heights.

**Figure 5 sensors-23-03525-f005:**
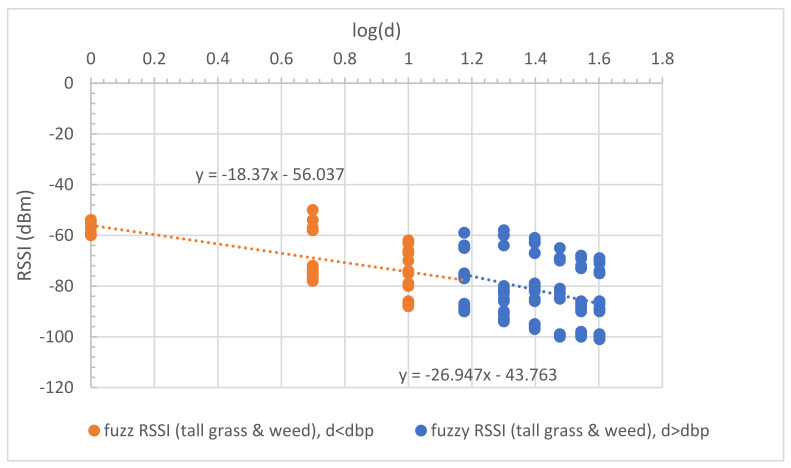
Uncertainty in observed RSSI and distance of tall grass and weed, for all antenna heights.

**Figure 6 sensors-23-03525-f006:**
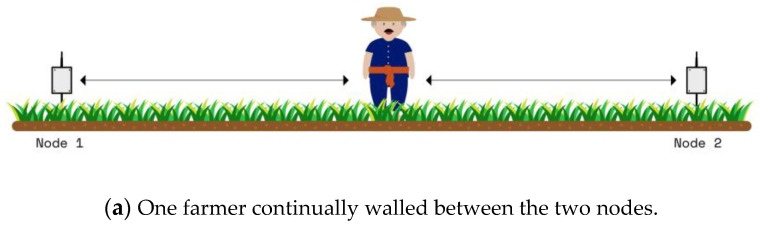
Blockage scenarios between the two nodes.

**Figure 7 sensors-23-03525-f007:**
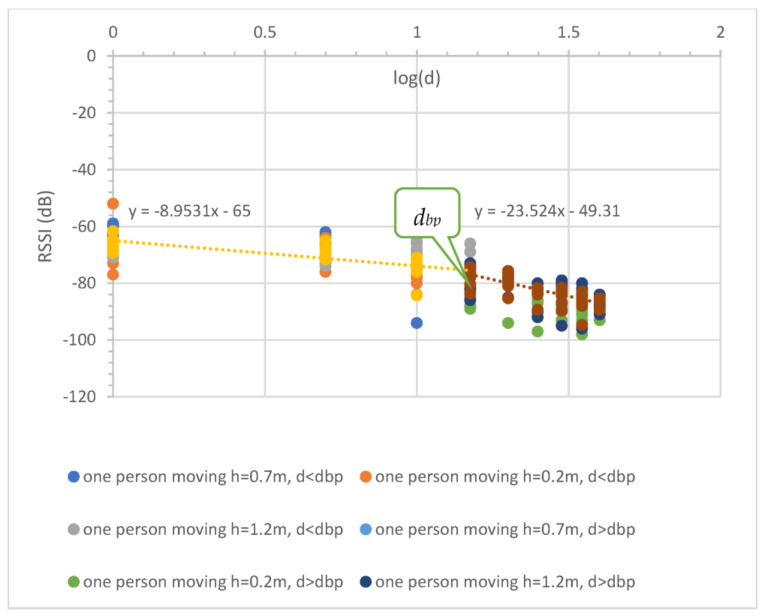
Uncertainty in observed RSSI and distance with one person (farmer) moving, for all antenna heights.

**Figure 8 sensors-23-03525-f008:**
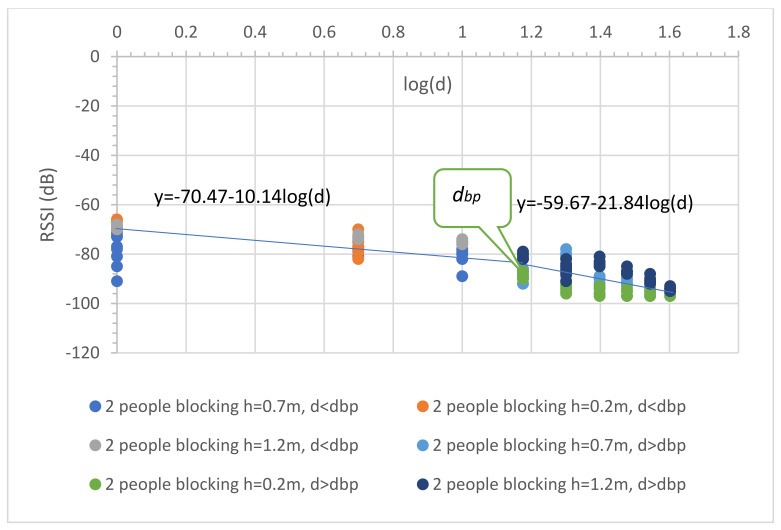
Uncertainty in observed RSSI and distance with two people (farmers) blocking, for all antenna heights.

**Figure 9 sensors-23-03525-f009:**
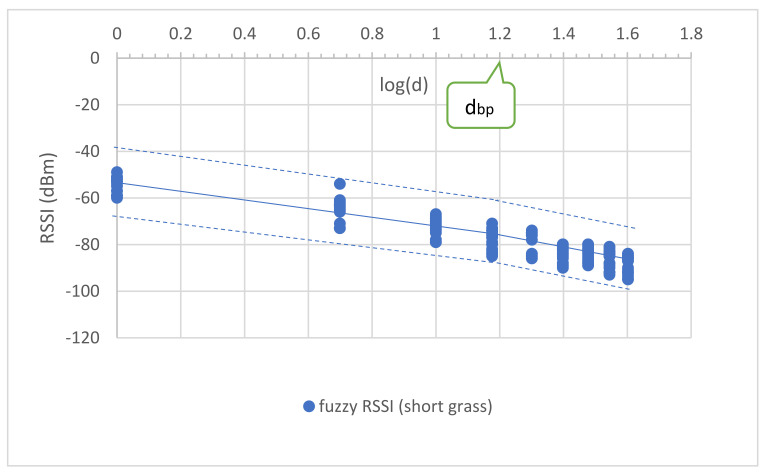
FLR curves for short grass with membership level 0.4.

**Figure 10 sensors-23-03525-f010:**
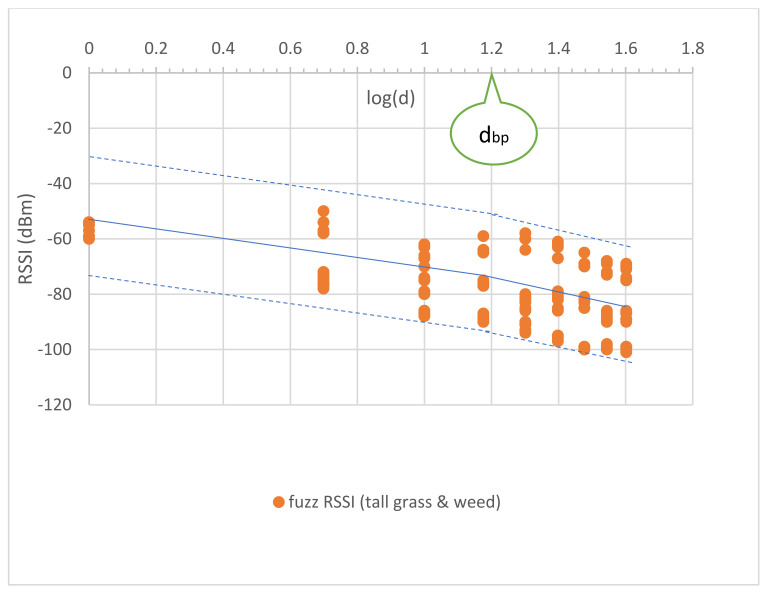
FLR curves for tall grass and weed with membership level 0.2.

**Figure 11 sensors-23-03525-f011:**
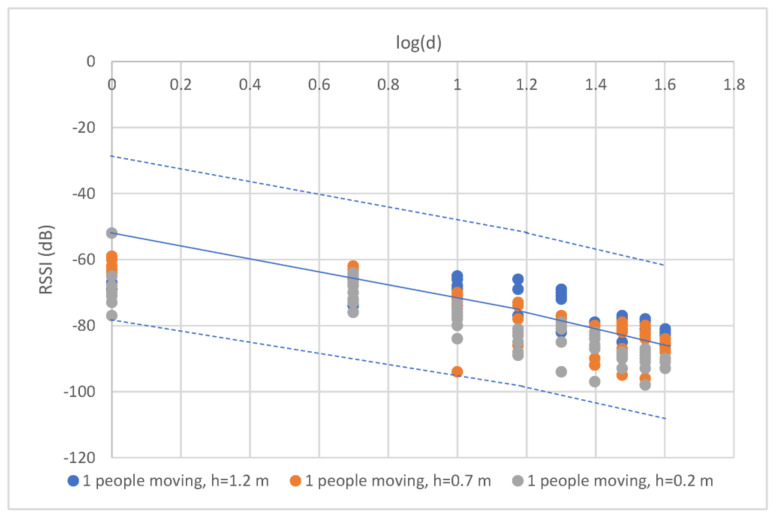
FLR curves for one person moving with membership level 0.0.

**Figure 12 sensors-23-03525-f012:**
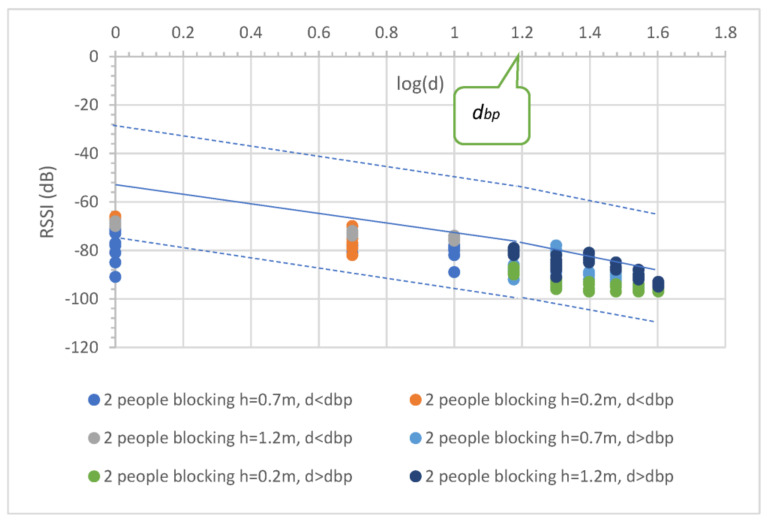
FLR curves for RSSI with two people blocking at membership level 0.0.

**Figure 13 sensors-23-03525-f013:**
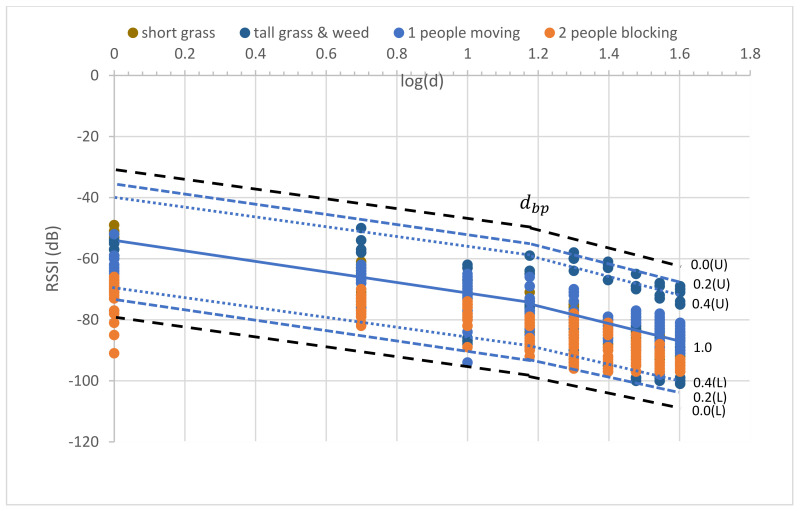
MBFLR for all fuzzy RSSI.

**Figure 14 sensors-23-03525-f014:**
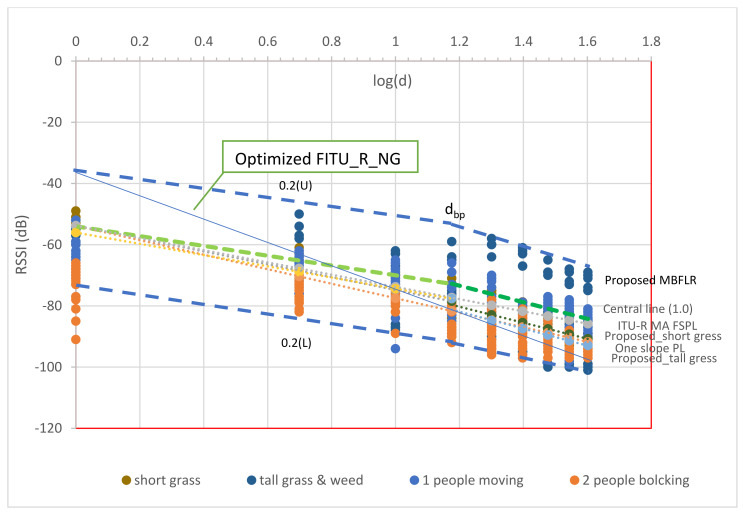
Comparisons between the proposed model and the other models.

**Table 1 sensors-23-03525-t001:** Summary of the equipment and measurement parameters.

Parameter	Value	Unit
	(min, max)	
Antenna gain (omni-directional)	2.3	dBi
Frequency	433	MHz
Bandwidth (BW)	125	kHz
Spreading Factor (SF)	SF7	
Pr (d_0_ = 1 m) (average)	−57	dBm
Output power	10	dBm
Coding Rate (CR)	4/5	
Antenna height (h_tx_, h_rx_)	(0.2, 0.7, 1.2)	m
Short grass height	(0.0, 0.3)	m
Tall grass height	(0.3, 0.5)	m
Breakpoint distance	15	m
Small-scale distance (λ/4)	0.4	m
Large-scale distance (Tx-Rx)	(1, 40)	m

**Table 2 sensors-23-03525-t002:** Statistical evaluation for each path loss model using RMSE.

Name	Models	RMSE		
		Short Grass	Tall Grass & Weed	People Blockage
Optimized FITU-R NG [17]	$AITU-R{\;}_{foliage}+PL_{PlaneEarth}=0.2\;f^{0.3}d^{0.6}+\left(40\log d\right)-\left(20\;log\;ht\right)-\left(20\;log\;hr\right)$	25.2	27.7	24.7
FITU-R MA FSPL [31]	$PL_{MA-ITU-R}\left(dB\right)=A_{M}\left(1-e^{-Rd/A_{M}}\right)+32.44+20\;log\;d+20\;log\;f$	4.3	10.9	8.1
One Slope PL LOS, 433 MHz [19]	$PL_{LOS\left(433MHz\right)}=PL_{0}+10nlog\left(d\right)$ (*n* = 2.37 for SF7 and BW 125 kHz)	5.7	11.8	8.3
Proposed dual-slope (Based on measurement)	- short grass: *RSSI*_1_*(d) =* −53.79 − 19.42*log(d); d < d_bp_* *RSSI_2_(d) =* −49.31 − 25.87*log(d); d > d_bp_* - tall grass: *RSSI*_1_*(d) =* −56.04 − 18.37*log(d); d < d_bp_* *RSSI*_2_*(d) =* −43.76 − 26.95*log(d); d > d_bp_*	4.6 4.4	11.4 10.9	8.0 7.8
Proposed (MBFLR)	*RSSI(d)* = [−55.4, −18.2] + [−16.2, 0.2]*log(d), d <*$\;d_{bp}$ *RSSI(d)* = [−42.9, −22.5] + [−26.8, 4.9]*log(d), d >* $d_{bp}$ - $\mu_{AI}$ *=* 0.4	0.8	1.2	2.6

## Data Availability

Not applicable.

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
