# Peer review of "Multi-Boundary Empirical Path Loss Model for 433 MHz WSN in Agriculture Areas Using Fuzzy Linear Regression"

_sensors, 2023, doi:10.3390/s23073525_

Round 1
Reviewer 1 Report
This paper investigates the ability of multiple boundary fuzzy linear regression (MBFLR) to establish uncertain relationships between variables relevant for path loss prediction in an agricultural tillage setting. It mainly applies multi boundary fuzzy linear regression MBFLR, which is used to establish the relationship between two or more variables. The main conclusion of this paper is that the proposed multi boundary MBFLR model provides the definition of upper and lower uncertainty bounds for the RSSI distance relation, which guarantees the stability of WSN in agricultural environments. Some have to be improved:
1. It does not take into account other environmental factors, such as humidity, temperature and wind speed, which also affect signal strength.
2. Further studies should be carried out on a larger scale and over a longer period of time to obtain more accurate path loss predictions under various environmental conditions.
3. Different types of vegetation cover are not taken into account.
4. The multiple boundary Fuzzy Linear regression (MBFLR) model for path loss prediction and its equations are not explained in detail.
5. It is not clear how exactly breakpoint distance optimization can help improve accuracy by accounting for uncertainties such as human or animal movement that may cause the network to be disconnected.
6. There is no explanation as to why certain open rural models only provide good predictions for open area environments without any blocking due to the first Fresnel zone area.
Author Response
"Please see the attachment"

Reviewer 2 Report
First of all, I want to congratulate the authors for their efforts in this manuscript. They have analyzed the changes in the RSSI when different vegetation or humans are between the nodes. Although the topic is not entirely novel, the following approach and results are interesting. There are some aspects that must be improved. Please find below a series of comments aimed at improving the quality of the manuscript.
1. The related work should be moved to a new subsection. The content of the section must be extended. The following reference is based on a similar approach, and it is necessary to compare the results (check similar results):
García, L., Parra, L., Jimenez, J. M., Parra, M., Lloret, J., Mauri, P. V., & Lorenz, P. (2021). Deployment strategies of soil monitoring WSN for precision agriculture irrigation scheduling in rural areas. Sensors, 21(5), 1693.
2. In the related work, split the information into different paragraphs.
3. Consider a different aspect for Figure 2 to ensure that it can be easily read when it is printed. Avoid using dark colours.
4. At the beginning of a section which is divided into a subsection, add a short paragraph detailing the content of the different subsections.
5. In the discussion, add a comparison with existing literature. Provide here a comparison with the results of papers included in the related work. It is recommended the use of a comparative table. In the same subsection, add the main limitations of the connected texts. If possible, include the discussion as a new section after the results.
6. Add future work in a new paragraph in the conclusions.
7. Minor issues:
Check the format. Many problems include the incorrect use of capital letters in Figures and Table. There are also problems with the paragraphs' line spacing.
Some figures are too big. Please consider reshaping the figures to reduce their size. It is unacceptable that there are only two graphics on some pages. This is making the paper too long.
Avoid using agriculture as a keyword since it is already used in the title.
Author Response
"Please see the revise version"

Reviewer 3 Report
I really appreciate the style of presentation of this paper, authors need to incorporate the underneath mentioned points for a better and possible publication by the journal. I therefore, recommend for revision.
1. The abstract should be had the main achievement and emphasise the novelty, so please rewrite it considering these points.
2. What is the motivation behind conducting the research. It is recom-
mended to highlight novelty of the work.
3. The critical literature review should be presented to indicate the draw-
backs of existed approaches, then, well define the main stream of research
direction, how did those previous studies performed, which problem still
requires to be solved and why is the proposed approach suitable to be
used to solve the critical problem?
4. Inappropriate methodology for answering your hypothesis, mention methodology in abstract.
5. Regarding English language the text must be clearly improved. There are
few of typos and English structural mistakes. The paper must be proof-
read by native English-speaking persons seriously.
6. Key results should be explained more elaboratively.
7. Make comparisons with previous work in the field and include articles that
have a clear statement of the improvements made to justify publication.
8. The conclusion section must be extracted and authors should use important outcomes that they obtained from the results section. Also discuss
how this research can lead to future enhancements.
Author Response
"Please see the revise version"

Round 2
Reviewer 1 Report
The modified part had better show in highlights.
Reviewer 2 Report
The authors have addressed the comments.